# Electromagnetic-guided versus endoscopic placement of nasoenteral feeding tubes: protocol for a systematic review and meta-analysis

Zheng Jin [ID] , Yaping Wei, Guofan Shen, Xiaofeng Zhang [ID]

► Prepublication history and additional materials for this paper is available online. To view these files, please visit the journal online (http://dx.doi.org/10.1136/bmjopen-2020-044637).

ZJ, YW and GS contributed equally.

Affiliated Hangzhou First People's Hospital, Zhejiang University School of Medicine, Hangzhou, Zhejiang, China

**Correspondence to**
Dr Xiaofeng Zhang;
gangdaoersan@163.com

## ABSTRACT

**Introduction** Current evidence supporting the utility of electromagnetic (EM)-guided method as the preferred technique for nasoenteral feeding tube placement is limited. We plan to provide a meta-analysis to compare the performance of EM-guided versus endoscopic placement.

**Methods and analysis** Randomised controlled trials evaluating EM-guided versus endoscopic placement will be searched in MEDLINE, EMBASE and CENTRAL from database inception to 30 September 2020. Data on study design, participant characteristics, intervention details and outcomes will be extracted. Primary outcomes to be assessed are complications. Secondary outcomes include procedure success rate, total procedure time, patient recommendation, length of hospital stay and mortality. Study quality will be assessed using the Cochrane risk of bias tool. Data will be combined with a random effects model. The results will be presented as a risk ratio for dichotomous data and weighted mean difference for continuous data. Publication bias will be visualised using funnel plots. We will quantify the effect of potential effect modifiers by meta-regression if appropriate. The quality of evidence will be evaluated according to the Grading of Recommendations Assessment, Development and Evaluation framework.

**Ethics and dissemination** This study will not use primary data, and therefore formal ethical approval is not required. The findings will be disseminated through peer-reviewed journals and committee conferences.

**PROSPERO registration number** CRD42020172427.

## Strengths and limitations of this study

► This systematic review and meta-analysis will update the existing clinical evidence of placement of nasoenteral feeding tubes.
► The thorough and transparent methodological approach undertaken will minimise the risk of possible biases. Quality of evidence will be assessed to provide confidence in the effect estimates.
► Multiple methods for investigating heterogeneity can inform intervention and study design, but contingent on the number and size of available studies.
► Common to any meta-analysis, some heterogeneity across and within studies may exist.

## INTRODUCTION

Malnutrition and inability to eat are conditions often encountered in inpatients. For such patients, enteral nutrition is considered to be superior to parenteral nutrition since it reduces complications, improves patient outcome and is more cheaper.[1 2]

It is common practice to place a nasoenteral feeding tube for enteral nutrition in patients who are intolerant of intragastric nutrition.[3] Endoscopic (ENDO) technique is typically used but requires patient transportation between wards, preprocedural fasting and radiological confirmation of the tube's position. Since first reported by Phang in 2006,[4] electromagnetic (EM)-guided technique has been increasingly used for nasoenteral feeding tube placement. With increasing availability and familiarity with this technique, several randomised controlled trials (RCTs)[5–8] have compared EM-guided versus ENDO technique. EM and ENDO techniques have been compared in only one systematic review until now.[9] That review by Gerritsen *et al* involving only one relevant RCT (66 patients) concluded that the efficacy and safety of the two techniques did not differ significantly, but EM offered advantages in logistics. In view of several new RCTs published, we have planned to pool the evidence to further evaluate the performance of EM versus ENDO.

## METHODS

The review will be performed according to the recommendations specified in the Cochrane Handbook for Intervention Reviews.[10] The reporting of the review will follow the Preferred Reporting Items for Systematic Reviews and Meta-Analyses (PRISMA) statement.[11]

## Criteria for considering studies for this review

Eligibility criteria are established in terms of the Population-Intervention-Comparison-Outcome-Study design framework. Studies will be selected according to the following criteria.

### Participants

Included studies will involve adult patients with an indication for enteral nutrition via a nasoenteral feeding tube, as indicated by the treating physician and/or consulting dietitian. Both critically ill and non-critically ill patients will be included. Exclusion criteria will be high suspicion of stenosis of the upper gastrointestinal tract, oesophageal varices, signs of upper gastrointestinal haemorrhage and pregnancy.

### Interventions/comparison

The intervention comparisons are EM versus ENDO. Studies with additional prokinetic agent administration will also be included.

### Outcomes

Primary outcome: complications.

There are six secondary outcomes: procedure success rate (defined as the percentage of successful tube placement in the desired location as determined), total procedure time, patient recommendation, length of hospital stay and mortality.

### Study design

Only RCTs (including factorial, cross-over, sequential design, and so on) will be included. An abstract with sufficient data will also be considered. We will only include studies that are presented in English language due to constraints in translational resources.

Studies will be excluded if it meets at least one of the following criteria: (1) observational studies including case–control and cohort studies; (2) single-arm studies; (3) case reports, reviews, editorials and letters to editor; (4) duplicate studies, in vitro studies or animal studies.

## Search methods for identification of studies

### Electronic searches

Two investigators (ZJ and YW) will independently search MEDLINE, EMBASE and CENTRAL for all entries through 30 September 2020 using the following search terms: 'Cortrak', 'electromagnetic', 'endoscopic', 'nasoenteral, or post-pyloric' and 'tube(s), feeding, or nutrition'. The search strategies (online supplemental appendix 1) will be decided on after a discussion among all reviewers. We will assess eligibility of the retrieved articles by title and abstract using predetermined inclusion criteria. If this information is insufficient for eligibility assessment, we will review the full article. If any up-to-date evidence is published during the review period, we will evaluate the eligibility of each study and consider its addition to the analysis. To further increase the robustness of the literature search, a manual recursive search of the Web of Science for a list of studies that cite the included studies and studies that the included studies cited will be carried out to identify other potentially relevant articles.

## Data collection and analysis

### Selection of studies

Decisions about study inclusion and exclusion will be made independently by two investigators (ZJ and YW). Disagreements will be resolved by consensus after a mutual discussion. The details of the study selection procedure are shown in a PRISMA flow chart (figure 1).

### Data extraction and management

Two investigators (YW and GS) will independently extract the appropriate data onto standardised extraction forms. The following data will be extracted from included trials: author, year of publication, country of origin, number of centres, participating operators, patient demographics, indications for enteral nutrition and study outcomes. When necessary data are not included in the published papers, the first or corresponding authors will be contacted for additional information. If there is no reply, we will analyse only the available data.

### Assessment of risk of bias in included studies

We will assign two independent investigators (YW and GS) to appraise methodological quality of the included trials with the Cochrane Collaboration's tool for assessing risk of bias.[12] The tool appraises existence of selection bias by assessing methods of randomisation and allocation concealment, performance and detection of biases by checking blinding of personnel and outcome assessment, and attrition and reporting bias by evaluating incomplete and selective data reporting. Each of the items is assigned a judgement of high, low or unclear risk.

### Data synthesis

Risk ratios will be calculated for categorical variables. Weighted mean differences will be calculated for continuous variables. Medians will be used if means are not available and SDs will be calculated or imputed when possible.[13] Owning to the assumption of inherently various study scenarios and study populations, a random effects model for all analyses will be assumed. Heterogeneity among studies will be assessed by calculating the $I^2$ statistics whereby $I^2<25\%$ indicates no heterogeneity, $25\%\leq I^2<50\%$ indicates mild heterogeneity, $50\%\leq I^2<75\%$ indicates moderate heterogeneity and $I^2 \geq75\%$ indicates strong heterogeneity.[14] We had planned that if sufficient studies ($\geq10$) are included in the analysis of primary outcomes, we would construct funnel plots to evaluate publication bias.[15] Egger's test for low sample size with a significance level of 0.05, and Begg and Mazumdar's test for high sample volume at a significance level of 0.1.[16] All statistical analyses will be performed using Review Manager V.5.4.1 (The Cochrane Collaboration, The Nordic Cochrane Centre, Copenhagen, Denmark).

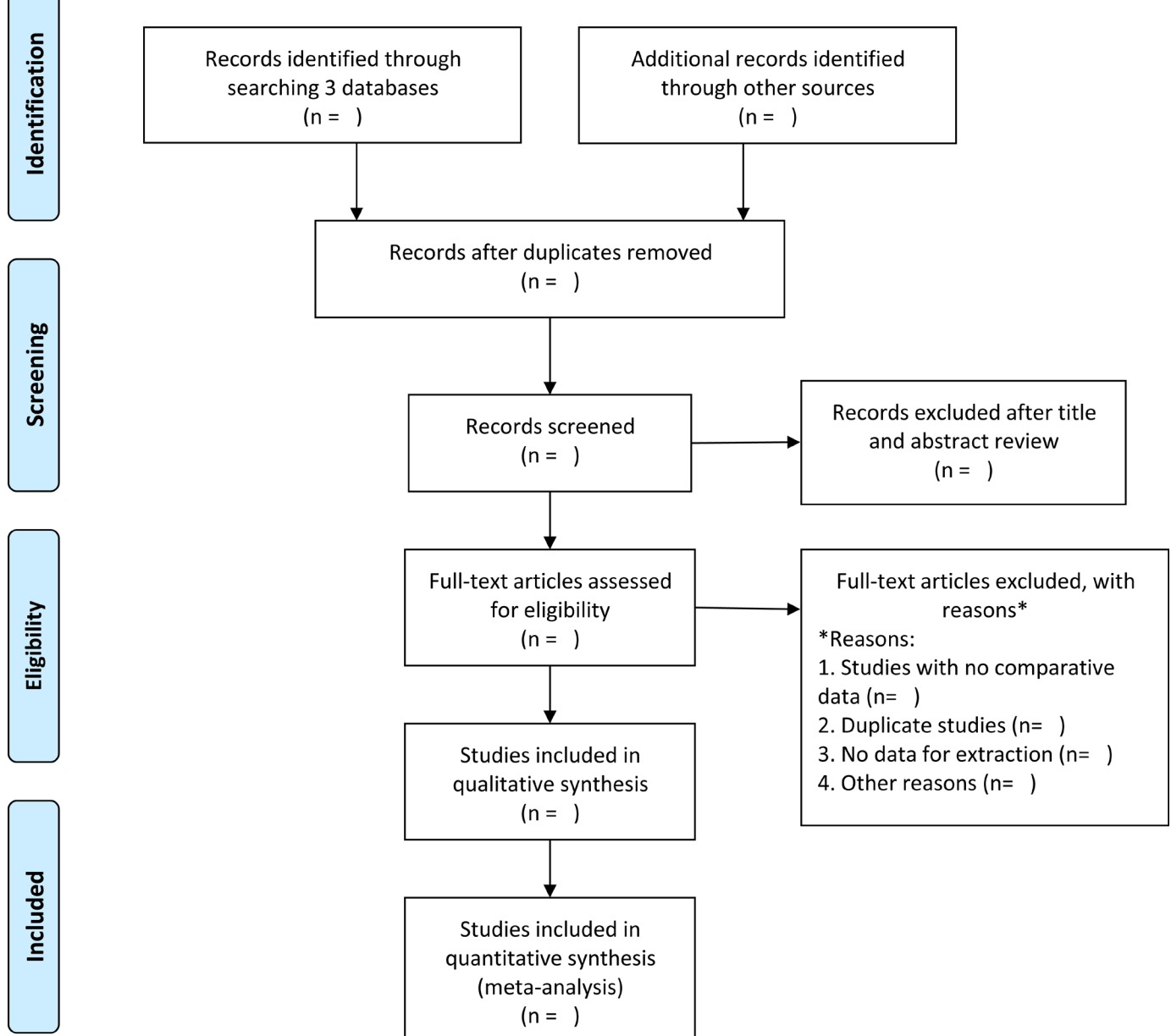

**Figure 1** Flow diagram of the study selection process.

## Meta-regression analysis

A random effects meta-regression analysis will be conducted to elucidate the impact of core set covariates (mean age, the proportion of males, sample size, research year and risk of bias) on treatment effects.

## Subgroup analyses

In the case of possible strong heterogeneity, we will explore the possible sources using subgroup analyses. It is reported that a higher body mass index (BMI) was associated with a successful placement for EM.[5 7] So, subgroup analyses will be carried out based on geographical location, study setting, the BMI level of patients with the validated method (ENDO group), prior altered upper gastrointestinal anatomy (with vs without) and patient population (critically ill vs non-critically ill patients). For

those subgroups with only one study included, subgroup analyses will not be performed.

## Sensitivity analysis

A sensitivity analysis will be conducted to determine whether our results are robust. We will exclude the studies with high risk for bias from the summary analysis and analyse them again to assess the impact of these studies on the results.

## Summary of evidence

We will summarise the quality of evidence using the Grading of Recommendations Assessment, Development and Evaluation approach and present 'Summary of findings' tables. Outcomes will be divided into critical (complications, procedure success rate, length of hospital

stay and mortality) and important (total procedure time and patient recommendation) outcomes.

## Patient and public involvement

Because the collected data within this systematic review and meta-analysis originate from previously published studies, patients and the general public were not involved in the development of the research question or choice of outcome measures that we wanted to assess.

## DISCUSSION

Conventional methods for the placement of nasoenteral feeding tubes include blind, fluoroscopic and ENDO methods.[17 18] The gold standard for the placement of nasoenteral feeding tube is the ENDO technique, which has success rates above 90%.[19 20] Bedside EM-guided tube placement can be performed in recent years. This has several potential advantages compared with ENDO placement because only one trained nurse and less equipment are needed.[21 22] But current evidence supporting the utility of EM-guided method as the preferred technique is limited. We therefore propose an updated meta-analysis to pool the evidence to further evaluate the performance of EM versus ENDO. The results of this study will influence the decision-making for the patients who have malnutrition or inability to eat, assist in future guideline development and guide future research endeavours.

## ETHICS AND DISSEMINATION

Ethics approval and patient's written informed consent will not be required because all analyses in the present study will be performed based on data from published studies. We will disseminate the findings of our work through conference presentations and a peer-reviewed publication.

**Contributors** ZJ and XZ conceived and designed this study. ZJ and YW searched and selected the studies. YW and GS extracted the essential information. YW and GS assessed the risk of bias. GS and ZJ performed the statistical analyses. ZJ and YW interpreted the pooled results. ZJ, YW and GS drafted the manuscript. All authors approved the manuscript to be considered for publication.

**Funding** The authors have not declared a specific grant for this research from any funding agency in the public, commercial or not-for-profit sectors.

**Competing interests** None declared.

**Patient consent for publication** Not required.

**Provenance and peer review** Not commissioned; externally peer reviewed.

**ORCID iDs**
Zheng Jin http://orcid.org/0000-0001-8319-1794
Xiaofeng Zhang http://orcid.org/0000-0003-3940-2982

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
