## [Reviewer comments · BMJ Open]

ARTICLE DETAILS

TITLE (PROVISIONAL)	Electromagnetic-guided versus endoscopic placement of nasoenteral feeding tubes: protocol for a systematic review and meta-analysis
AUTHORS	Jin, Zheng; Wei, Yaping; Shen, Guofan; Zhang, Xiaofeng

VERSION 1 – REVIEW

REVIEWER	Dr. Rostam Jalai Kermanshah university of medical sciences
REVIEW RETURNED	15-Oct-2020

GENERAL COMMENTS	Thanks for giving me the opportunity to judge this article. It is a good study, but the following needs to be considered and corrected: 1. Since randomized clinical trial (RCT) studies are intended for use in the study, please mention studies such as other trial studies in the study design section, including clinical trial, factorial, cross-over, sequential. observational studies including case-control and cohort studies, in the Excluded Studies section.2. Since this study is a protocol, you should mention both types of diffusion bias evaluation tests, Eger test for low sample size with a significance level of 0.05 and begg and Mazumdar test for high sample volume at a significance level of 0.1.3. Since the heterogeneity of the studies may be high, it is necessary to use the meta-regression test to examine the heterogeneity factors, so before the subgroup analysis section, add a section for meta-regression and heterogeneity factors such as sample size and research year
--

REVIEWER	Hiraku Tsujimoto Hyogo Prefectural Amagasaki General Medical Center, Japan
REVIEW RETURNED	01-Nov-2020

GENERAL COMMENTS	This is a systematic review protocol for Electromagnetic-guided versus endoscopic placement of gastroenteric feeding tubes. The topic is important for patients' safety. However, the methods of a systematic review (Cochrane handbook 2011) they planned to use are out of date. They should comprehensively update the methods. I think it would be a better study to have a more solid plan in place before running a search. <Major comments> #1 Outcomes Please see Cochrane handbook Version 6.1, 2020.
--

	Please change the primary and secondary outcomes into patients' relevant outcomes. "In general, systematic reviews should aim to include outcomes that are likely to be meaningful to the intended users and recipients of the reviewed evidence." If I have a choice of how to put in an enteric tube, I choose it based on the data that it is a method that, for whatever reason, does not result in serious complications, such as gastrointestinal perforation or aspiration of nutrients into the lungs. I think it is only when there was no difference in those outcomes that I would refer to the evaluation of other outcomes. #2 Outcomes Please limit the number of outcome domains to less than seven. Please see the Cochrane handbook Version 6.1, 2020. #2 Search Search using Google scholar is generally not reproducible and not suitable for a formal systematic review. I think it's a good idea to use it as an adjunct or additional citation search, but please leave it out of the abstract. #3 Search Search the Web of Science for a list of studies that cite the included studies (citation search) and screen studies that the included studies cited. These are essential in a systematic review. Please see "Cochrane handbook Chapter 4: Searching for and selecting studies" #4 GRADE Please write that you will assess the quality of the evidence-based on GRADE. It is the current standard. Please see Cochrane handbook Version 6.1, 2020. #4 Online supplementary appendix 1. PubMed-MEDLINE search strategy Please remove "randomized controlled trial"[pt] OR "controlled clinical trial"[pt] OR "randomized"[tiab] OR "randomised"[tiab] OR "randomly"[tiab]. Only use validated RCT filters. Please see Cochrane handbook Version 6.1, 2020 Chapter 4: Searching for and selecting studies. #5 Search strategies Using wild cards is good, but I think the search is not comprehensive enough. A comprehensive search is a basis on which a systematic review can be a study, not just a literature review. Please add a few more synonyms to each concept. Also, please include at least one control word for each concept, such as MeSH for MEDLINE and MTHREE for EMBASE. #5 Search strategies Please provide all the search formula that the authors plan to use. This is a research protocol paper. #6 "Subgroup analyses will be carried out based on geographical location, study setting, the BMI level of patients with the validated method (ENDO group), prior altered upper gastrointestinal anatomy (with vs. without), patient population (critically ill vs. non-critically ill patients). For those subgroups with only 1 study included, subgroup analyses will not be performed."
--	--

	Please describe why these subgroups are important to divide in the “introduction”, citing previous researches. #7 Remove claims such as "first time". I don't think it has much to do with science who first planned the protocols for open data analysis. What matters, I think, is how they do it. #8 Since the authors are publishing the protocol as a paper, they should write on the strength of its methodological soundness. #9 Cost analysis Describe how costs are analyzed in a way that leads to clinical decision-making. For example, what is the cost to whom? <Minor comments> # Please remove “Meta-analysis will be performed using RevMan V.5.3 statistical software” in the abstract. The software is not important in the abstract. # Update RevMan to the latest version RevMan 5.4.1 or later. #Please update your PROSPERO record. In the record, the anticipated completion date is 20 April 2020.
--	--

VERSION 1 – AUTHOR RESPONSE

Reviewer: 1

1. Since randomized clinical trial (RCT) studies are intended for use in the study, please mention studies such as other trial studies in the study design section, including clinical trial, factorial, cross-over, sequential. observational studies including case-control and cohort studies, in the Excluded Studies section.

Response: Thanks for your advice. Modifications have been made.

2. Since this study is a protocol, you should mention both types of diffusion bias evaluation tests, Egger test for low sample size with a significance level of 0.05 and Begg and Mazumdar test for high sample volume at a significance level of 0.1.

Response: Thanks for your advice. Modifications have been made.

3. Since the heterogeneity of the studies may be high, it is necessary to use the meta-regression test to examine the heterogeneity factors, so before the subgroup analysis section, add a section for meta-regression and heterogeneity factors such as sample size and research year

Response: Thanks for your advice. Modifications have been made.

Reviewer: 2

Comments to the Author

This is a systematic review protocol for Electromagnetic-guided versus endoscopic placement of gastroenteric feeding tubes. The topic is important for patients' safety. However, the methods of a

systematic review (Cochrane handbook 2011) they planned to use are out of date. They should comprehensively update the methods. I think it would be a better study to have a more solid plan in place before running a search.

Response: Thanks for your advice. We have updated our methods according to Cochrane handbook Version 6.1, 2020.

#1 Outcomes

Please see Cochrane handbook Version 6.1, 2020.

Please change the primary and secondary outcomes into patients' relevant outcomes.

"In general, systematic reviews should aim to include outcomes that are likely to be meaningful to the intended users and recipients of the reviewed evidence."

If I have a choice of how to put in an enteric tube, I choose it based on the data that it is a method that, for whatever reason, does not result in serious complications, such as gastrointestinal perforation or aspiration of nutrients into the lungs. I think it is only when there was no difference in those outcomes that I would refer to the evaluation of other outcomes.

Response: Thanks for your advice. Complication has been selected for our primary outcome.

#2 Outcomes

Please limit the number of outcome domains to less than seven. Please see the Cochrane handbook Version 6.1, 2020.

Response: Thanks for your advice. Modifications have been made.

#2 Search

Search using Google scholar is generally not reproducible and not suitable for a formal systematic review.

I think it's a good idea to use it as an adjunct or additional citation search, but please leave it out of the abstract.

Response: Thanks for your advice. Modifications have been made.

#3 Search

Search the Web of Science for a list of studies that cite the included studies (citation search) and screen studies that the included studies cited. These are essential in a systematic review. Please see "Cochrane handbook Chapter 4: Searching for and selecting studies"

Response: Thanks for your advice. Modifications have been made in Electronic searches section.

#4 GRADE

Please write that you will assess the quality of the evidence based on GRADE. It is the current standard. Please see Cochrane handbook Version 6.1, 2020.

Response: Thanks for your advice. Modifications have been made in Summary of evidence section.

#4 Online supplementary appendix 1. PubMed-MEDLINE search strategy

Please remove "randomized controlled trial"[pt] OR "controlled clinical trial"[pt] OR "randomized"[tiab] OR "randomised"[tiab] OR "randomly"[tiab]. Only use validated RCT filters. Please see Cochrane handbook Version 6.1, 2020 Chapter 4: Searching for and selecting studies.

Response: Thanks for your advice. Filters to identify randomized trials have been developed specifically for MEDLINE and Embase. CENTRAL, however, aims to contain only reports with study designs possibly relevant for inclusion in Cochrane Reviews, so searches of CENTRAL should not use a trials 'filter' or be limited to human studies. (Cochrane handbook Version 6.1, 2020 Chapter 4: Searching for and selecting studies.) Modifications have been made in Online supplementary appendix 1.

#5 Search strategies

Using wild cards is good, but I think the search is not comprehensive enough. A comprehensive search is a basis on which a systematic review can be a study, not just a literature review. Please add a few more synonyms to each concept. Also, please include at least one control word for each concept, such as MeSH for MEDLINE and MTHREE for EMBASE.

Response: Thanks for your advice. Modifications have been made.

#5 Search strategies

Please provide all the search formula that the authors plan to use. This is a research protocol paper.

Response: Thanks for your advice. Search formulas have been provided in Online supplementary appendix 1 now.

#6 “Subgroup analyses will be carried out based on geographical location, study setting, the BMI level of patients with the validated method (ENDO group), prior altered upper gastrointestinal anatomy (with vs. without), patient population (critically ill vs. non-critically ill patients). For those subgroups with only 1 study included, subgroup analyses will not be performed.” Please describe why these subgroups are important to divide in the “introduction”, citing previous researches.

Response: Thanks for your advice. We chose these factors according to previous researches and clinical practice. It is reported that a higher body mass index (BMI) were associated with a successful placement for EM. (1. Gao X, Zhang L, Zhao J, et al.. Bedside electromagnetic-guided placement of nasoenteral feeding tubes among critically ill patients: A single-centre randomized controlled trial. J CRIT CARE. 2018; 48: 216-21.

Holzinger U, Brunner R, Miehsler W, et al.. Jejunal tube placement in critically ill patients: A prospective, randomized trial comparing the endoscopic technique with the electromagnetically visualized method. CRIT CARE MED. 2011; 39: 73-7) Modifications have been made in Subgroup analyses section.

#7 Remove claims such as "first time". I don't think it has much to do with science who first planned the protocols for open data analysis. What matters, I think, is how they do it.

Response: Thanks for your advice. Modifications have been made.

#8 Since the authors are publishing the protocol as a paper, they should write on the strength of its methodological soundness.

Response: Thanks for your advice. Modifications have been made.

#9 Cost analysis

Describe how costs are analyzed in a way that leads to clinical decision-making. For example, what is the cost to whom?

Response: Thanks for your advice. Total costs included the costs for tube placement procedure, complications and therapeutic interventions. Currency units were variably used euro and dollar across the studies. Standard mean differences will be calculated for total costs based on different monetary units. Modifications have been made.

Please remove “Meta-analysis will be performed using RevMan V.5.3 statistical software” in the abstract. The software is not important in the abstract.

Response: Thanks for your advice. Modifications have been made.

Update RevMan to the latest version RevMan 5.4.1 or later.

Response: Thanks for your advice. We have updated the software.

#Please update your PROSPERO record. In the record, the anticipated completion date is 20 April 2020.

Response: Thanks for your advice. We have update the record. Now it is waiting their system approval.

VERSION 2 – REVIEW

REVIEWER	Hiraku Tsujimoto Kyoto University, Japan
REVIEW RETURNED	29-Dec-2020

GENERAL COMMENTS	#1 page 9, line 3 “If there are no data on any of the primary or secondary outcomes, those studies will be excluded from the meta-analyses.” Since this is a systematic review, we should not incorporate data availability into inclusion/exclusion criteria. It may introduce bias of the result. The authors should report them independently as INCLUDED studies without adequate data. #2 page 11, line 5, in Sensitivity analysis “We will carry out a sensitivity analysis by systematically removing every study and checking the pooled results for the remaining studies to see if there is any significant change in test performance.” I think that a sensitivity analysis should be based on some hypothesis. This should always be related to the vulnerability of the conclusions of the review, and not necessarily known at the protocol stage. It seems desirable to have a flexible format that allows for additional analysis after the review. #3 page 11, line 9, in Summary of evidence “We will summarise the quality of evidence using the GRADE (Grading of Recommendations Assessment, Development and Evaluation) approach and present ‘Summary of findings’ tables.” Please write in your plan which outcome domains, called the main outcomes (which is a separate concept from the primary/secondary outcomes), you will report in the SoF. #4 Cost analysis According to the Cochrane handbook that the authors cited, “Two optional methodological frameworks have therefore been developed for incorporating economic evidence into reviews. The methodological and practical implications of each approach should be considered carefully at an early stage of planning the protocol for a systematic review. The two methodological frameworks are: (1) integrated full systematic review of economic evidence; and (2) brief economic commentary.” In this case it would be “a brief economic commentary”. According to the handbook, please mention (1) the economic burden of the health condition (i.e. the ‘cost of illness’); (2) potential impacts of intervention(s) on resource use (costs); and (3) general issues of intervention costs and cost-effectiveness that
--

	are relevant for the readers of the review to consider in the Background section. Since this is not a Cochrane review, a brief description is OK. #5 Cost analysis It would be very difficult to follow all of them, but please read Chapter 20: Economic evidence, and if there is anything that the authors can do at this stage, please incorporate it.
--	---

VERSION 2 – AUTHOR RESPONSE

Reviewer: 2

1. page 9, line 3 “If there are no data on any of the primary or secondary outcomes, those studies will be excluded from the meta-analyses.” Since this is a systematic review, we should not incorporate data availability into inclusion/exclusion criteria. It may introduce bias of the result. The authors should report them independently as INCLUDED studies without adequate data.

Response: Thanks for your advice. That sentence has been deleted.

2. page 11, line 5, in Sensitivity analysis “We will carry out a sensitivity analysis by systematically removing every study and checking the pooled results for the remaining studies to see if there is any significant change in test performance.” I think that a sensitivity analysis should be based on some hypothesis. This should always be related to the vulnerability of the conclusions of the review, and not necessarily known at the protocol stage. It seems desirable to have a flexible format that allows for additional analysis after the review..

Response: Thanks for your advice. According to the Cochrane handbook, some sensitivity analyses can be pre-specified in the study protocol, but many issues suitable for sensitivity analysis are only identified during the review process where the individual peculiarities of the studies under investigation are identified. (Cochraene handbook Version 6.1, 10.14 Sensitivity analyses.) Modifications have been made.

3. page 11, line 9, in Summary of evidence “We will summarise the quality of evidence using the GRADE (Grading of Recommendations Assessment, Development and Evaluation) approach and present ‘Summary of findings’ tables.”

Please write in your plan which outcome domains, called the main outcomes (which is a separate concept from the primary/ secondary outcomes), you will report in the SoF.

Response: Thanks for your advice. According to the Cochrane handbook, to ensure production of optimally useful information, Cochrane Reviews begin by developing a review question and by listing all main outcomes that are important to patients and other decision makers. The GRADE approach to assessing the certainty of the evidence defines and operationalizes a rating process that helps separate outcomes into those that are critical, important or not important for decision making. (Cochraene handbook Version 6.1, 14.1.2 Selecting outcomes for ‘Summary of findings’ tables.) In our review, outcomes will be divided into critical (complications, procedure success rate, length of hospital stay, and mortality), and important (total procedure time, and patient recommendation) outcomes. Modifications have been made.

4. Cost analysis

According to the Cochrane handbook that the authors cited, “Two optional methodological frameworks have therefore been developed for incorporating economic evidence into reviews. The methodological and practical implications of each approach should be considered carefully at an early stage of planning the protocol for a systematic review. The two methodological frameworks are: (1) integrated full systematic review of economic evidence; and (2) brief economic commentary.” In this case it would be “a brief economic commentary”.

According to the handbook, please mention (1) the economic burden of the health condition (i.e. the ‘cost of illness’); (2) potential impacts of intervention(s) on resource use (costs); and (3) general issues of intervention costs and cost-effectiveness that are relevant for the readers of the review to consider in the Background section. Since this is not a Cochrane review, a brief description is OK.

Response: Thanks for your advice. According to the Cochrane handbook, whilst all reviews could have an economic component, an economic component might not always be necessary. (Cochrane handbook Version 6.1, 20.1.3 Criteria for prioritizing inclusion of economic evidence in a Cochrane Review.) Because of the limited information among studies, we have decided not to incorporate economic evidence into our review.

5. It would be very difficult to follow all of them, but please read Chapter 20: Economic evidence, and if there is anything that the authors can do at this stage, please incorporate it.

Response: Thanks for your advice. We have decided not to incorporate economic evidence into our review.

VERSION 3 – REVIEW

REVIEWER	Hiraku Tsujimoto Kyoto University
REVIEW RETURNED	04-Feb-2021

GENERAL COMMENTS	I think the manuscript improved and these authors will be able to conduct a good systematic review using this protocol.
---